# Phenotypic and Genotypic Characterization of Multidrug-Resistant *Enterobacter hormaechei* Carrying *qnrS* Gene Isolated from Chicken Feed in China

Zhengzheng Cao,[a,b] Luqing Cui,[a,b] Quan Liu,[a,b] Fangjia Liu,[a,b] Yue Zhao,[a,b] Kaixuan Guo,[a,b] Tianyu Hu,[a,b] Fan Zhang,[a,b] Xijing Sheng,[a,b] ⬤Xiangru Wang,[a] ⬤Zhong Peng,[a] ⬤Menghong Dai[a,b]

aThe Cooperative Innovation Center for Sustainable Pig Production, Huazhong Agricultural University, Wuhan, China
bMOA Key Laboratory of Food Safety Evaluation/National Reference Laboratory of Veterinary Drug Residue (HZAU), Huazhong Agricultural University, Wuhan, China

**ABSTRACT** Multidrug resistance (MDR) in Enterobacteriaceae including resistance to quinolones is rising worldwide. The plasmid-mediated quinolone resistance (PMQR) gene *qnrS* is prevalent in Enterobacteriaceae. However, the *qnrS* gene is rarely found in *Enterobacter hormaechei* (*E. hormaechei*). Here, we reported one multidrug resistant *E. hormaechei* strain M1 carrying the *qnrS1* and *bla*TEM-1 genes. This study was to analyze the characteristics of MDR *E. hormaechei* strain M1. The *E. hormaechei* strain M1 was identified as *Enterobacter cloacae* complex by biochemical assay and 16S rRNA sequencing. The whole genome was sequenced by the Oxford Nanopore method. Taxonomy of the *E. hormaechei* was based on multilocus sequence typing (MLST). The *qnrS* with the other antibiotic resistance genes were coexisted on IncF plasmid (pM1). Besides, the virulence factors associated with pathogenicity were also located on pM1. The *qnrS1* gene was located between insertion element IS2A (upstream) and transposition element ISKra4 (downstream). The comparison result of IncF plasmids revealed that they had a common plasmid backbone. Susceptibility experiment revealed that the *E. hormaechei* M1 showed extensive resistance to the clinical antimicrobials. The conjugation transfer was performed by filter membrane incubation method. The competition and plasmid stability assays suggested the host bacteria carrying *qnrS* had an energy burden. As far as we know, this is the first report that *E. hormaechei* carrying *qnrS* was isolated from chicken feed. The chicken feed and poultry products could serve as a vehicle for these MDR bacteria, which could transfer between animals and humans through the food chain. We need to pay close attention to the epidemiology of *E. hormaechei* and prevent their further dissemination.

**IMPORTANCE** *Enterobacter hormaechei* is an opportunistic pathogen. It can cause infections in humans and animals. Plasmid-mediated quinolone resistance (PMQR) gene *qnrS* can be transferred intergenus, which is leading to increase the quinolone resistance levels in Enterobacteriaceae. Chicken feed could serve as a vehicle for the MDR *E. hormaechei*. Therefore, antibiotic-resistance genes (ARGs) might be transferred to the intestinal flora after entering the gastrointestinal tract with the feed. Furthermore, antibiotic-resistant bacteria (ARB) were also excreted into environment with feces, posing a huge threat to public health. This requires us to monitor the ARB and antibiotic-resistant plasmids in the feed. Here, we demonstrated the characteristics of one MDR *E. hormaechei* isolate from chicken feed. The plasmid carrying the *qnrS* gene is a conjugative plasmid with transferability. The presence of plasmid carrying antibiotic-resistance genes requires the maintenance of antibiotic pressure. In addition, the *E. hormaechei* M1 belonged to new sequence type (ST). These data show the MDR *E. hormaechei* M1 is a novel strain that requires our further research.

**KEYWORDS** *Enterobacter hormaechei*, *qnrS*, chicken feed, IncF, multidrug resistance, transfer, gene transfer

Address correspondence to Menghong Dai, daimenghong@mail.hzau.edu.cn.

The authors declare no conflict of interest.

*E*nterobacter hormaechei is a member of the *E. cloacae* complex and has different subspecies (1). *E. hormaechei* is widespread in the environment, as a nosocomial pathogen causing neonatal bloodstream and urinary infections (2, 3). It also causes infections in animals. For example, *E. hormaechei* brought about respiratory disease for calves and septic arthritis in a green sea turtle (4, 5). Moreover, *E. hormaechei* strains that showed resistance to the major clinical antibiotics were isolated from the cloacae of poultry and a hand washing sink of a veterinary hospital (6, 7). QnrS, plasmid-mediated quinolone resistance (PMQR) gene, encodes for a protein of the pentapeptide repeat family that protects DNA gyrase and topoisomerase IV from quinolones inhibition (8).

PMQR genes are prevalent in Enterobacteriaceae, especially *Escherichia coli*, *Klebsiella*, *Salmonella*, and some species of *Enterobacter* (9–17). QnrS usually coexisted with multiple antibiotic resistance genes in the same plasmid to mediate multidrug resistance (MDR) (18–20). Currently, *E. hormaechei* carrying the *qnrS* gene has been reported (21). Moreover, our recent study showed that *E. coli* carrying the *qnrS* gene were isolated from chicken feed (22). However, the contamination of *E. hormaechei* in the feed has not been reported.

In this study, we aimed to analyze characteristics of *E. hormaechei* strain M1 carrying the *qnrS1* gene from the chicken feed, including the resistance level of *E. hormaechei* strain M1, transferability of plasmid, molecular traits of plasmid, as well as the genetic context of the *qnrS1* gene.

## RESULTS

**The identification of *E. hormaechei* M1 isolated from chicken feed.** The conventional biochemical tests were used to identify *E. hormaechei* M1. Through sugar and alcohol metabolism, amino acid and protein metabolism, carbon and nitrogen source utilization, and enzyme tests (Table 1), it was preliminarily determined to be *E. hormaechei*.

**Analysis of genome characteristics of *E. hormaechei* M1.** Sequencing results generated a complete chromosome of 4,658,400 bp and a plasmid 148,434 bp. The *oqxA*, *oqxB*, *bacA*, $bla_{ACT-25}$, and *fosA2* genes were located on chromosome. The *catA2*, *tet(D)*, *dfrA14*, *aph(3')-Ib*, *aph(6)-Id*, *qnrS1*, *sul2*, $bla_{TEM-1}$ genes coexisted in the same plasmid; the plasmid incompatibility group was IncF, it had two replicons IncFII and IncFIB.

**Characterization of plasmids.** The circular image and circular comparisons between pM1 and other reported similar IncF plasmids were completed using the BRIG tool (Fig. 1). Plasmids were included in the following order (inner to outer circles): p2-020038, pE70, pC44-01, pC4-001, and pM1. These plasmids harbored the IncFII and IncFIB replicons. However, the pM1 had unique genes, such as antibiotic resistance genes (*qnrS1* and $bla_{TEM-1}$), insertion sequences (InsD and IS2A), transposable elements (ISKra4 and tnpR), and pili operon (*traD*, *traF*, *traN*, *traU*, *traW*, *traT*, *traG*, *traH*, *trbB* and *trbC*). Meanwhile, these plasmids all harbored the *sul2* (sulfonamides), *dfrA14* (trimethoprim), *aph(3″)-Ib* (aminoglycosides), *aph(6)-Id* (aminoglycosides) resistance genes, and copper and silver heavy metal resistance genes. Moreover, the features were related to virulence, for example, copper and silver metal cation and tellurium ion efflux system, *sopA* and *sopB*, virulence-associated protein VagC, fimbrial operon, type IV pili and the *hok* toxic. The pM1 had a size of 148,434 bp with an average G+C content of 52%, and carried the *qnrS*1 gene that was located within the physical boundaries demarcated by insertion element IS2A (upstream) and transposition element ISKra4 (downstream). A genetic structure surrounding the *qnrS1* gene was detected from pM1(IS407-$bla_{TEM-1}$-IS2A-*qnrS1*-ISKra4-tnpR).

**Analysis of phylogenetic tree.** The *dnaA*, *fusA*, *gyrB*, *leuS*, *pyrG*, *rplB*, and *rpoB* 7 housekeeping genes were determined by multilocus sequence typing (MLST) (https://pubmlst.org/). The seven genes were concatenated to make a phylogenetic tree (Fig. 2). The 17 strains of *E. hormaechei* were divided into five different groups and distinguished by different colors. Sequence type (ST) and isolate source of the strain were listed in tree. The phylogenetic tree showed that the *E. hormaechei* M1 was divided into *Enterobacter hormaechei* subsp. *xiangfangensis* group. It was closely related to the evolution of *Enterobacter hormaechei* isolated from human.

**TABLE 1** Biochemical identification of *E. hormaechei* M1

| Test | Result[a] | Test | Result |
|---|---|---|---|
| Maltose | + | Citrate | + |
| Glucose | + | Indole | − |
| Lactose | − | Voges-Proskauer | − |
| Sucrose | + | Methyl red | − |
| Arabinose | + | Ornithine decarboxylase | + |
| Mannose | + | Malonate | + |
| Mannitol | + | D- Sorbitol | + |
| Dulcitol | − | D- Arabitol | − |
| α-D- Melibiose | + | Adonitol | − |

[a]+, positive; −, negative.

**Antimicrobials resistance of *E. hormaechei* M1 strain.** Susceptibility testing revealed that *E. hormaechei* M1 was resistant to most of the antimicrobials tested (Table 2). *E. hormaechei* M1 showed resistance to the following antimicrobials: β-lactams antibiotics (ampicillin, amoxicillin and cefalexin), quinolones (enrofloxacin), chloramphenicol, florfenicol, tetracyclines (oxytetracycline, tetracycline and doxycycline), sulfonamides (sulfadiazine), fosfomycin, compound medicines (amoxicillin/clavulanate potassium, trimethoprim-sulfamethoxazole). However, *E. hormaechei* M1 was sensitive to meropenem, gentamicin and polymyxin.

**The conjugation transfer efficiency and the characteristics of transconjugants resistance.** The conjugation transfer experiment showed that donor *E. hormaechei* M1 strain transferred the plasmid pM1 to the recipient *E. coli* C600 strain. Conjugation transfer efficiency was $4.6 \times 10^{-5} \pm 4.1 \times 10^{-5}$. The susceptibility results of the recipient strain

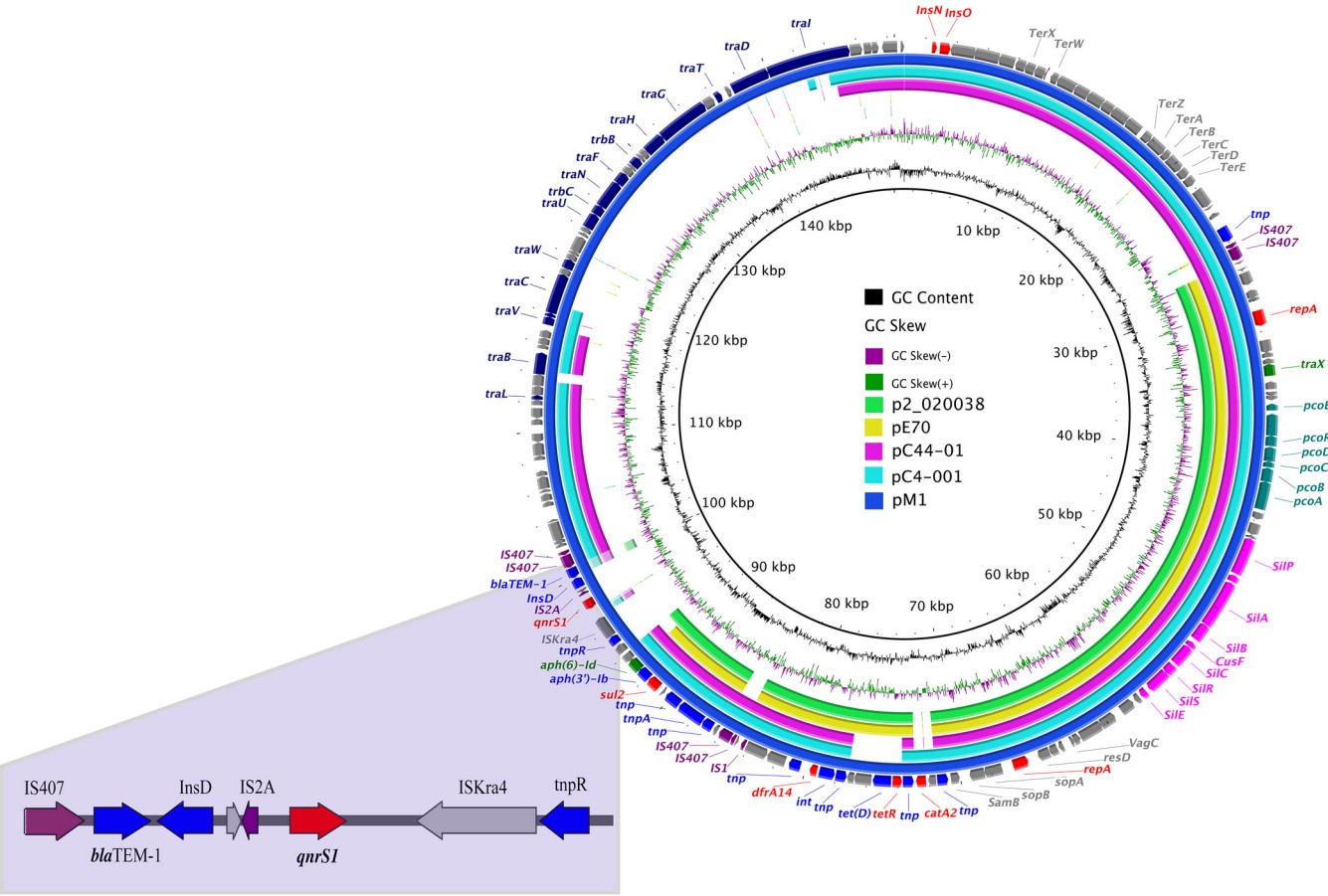

**FIG 1** The complete sequence of pM1 (the outer circle) was used as a reference plasmid. The circular maps were generated using the BRIG software, and plasmids were included in the following order (inner to outer circles): p2-020038 (GenBank ID: CP031723.1), pE70 (CP046273.1), pC44-01 (CP042567.1), pC4-001 (CP042541.1), and pM1 (CP090910).

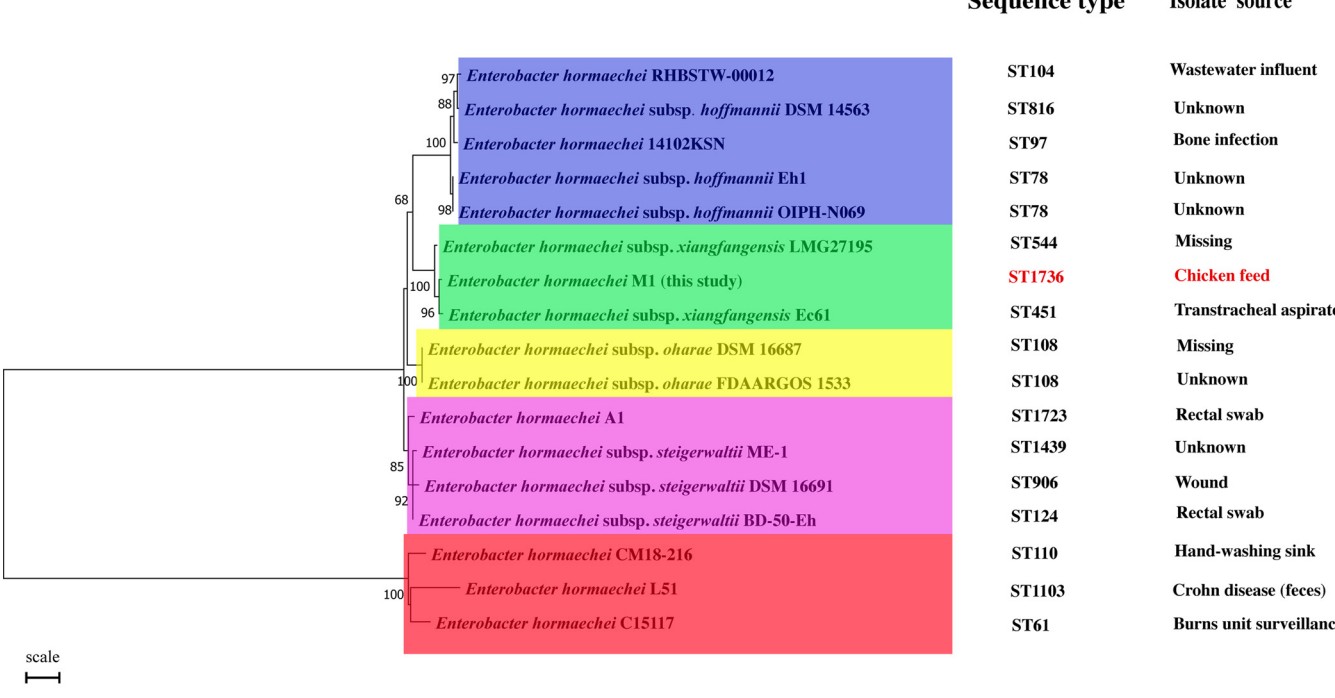

**FIG 2** The evolutionary history was inferred by using the maximum likelihood method and Hasegawa-Kishino-Yano model. The tree with the highest log likelihood (−7,997.87) is shown. The percentage of trees in which the associated taxa clustered together is shown next to the branches. Initial tree(s) for the heuristic search were obtained automatically by applying Neighbor-Join and BioNJ algorithms to a matrix of pairwise distances estimated using the maximum composite likelihood (MCL) approach, and then selecting the topology with superior log likelihood value. A discrete Gamma distribution was used to model evolutionary rate differences among sites (five categories [+G, parameter = 0.2152]). The tree is drawn to scale, with branch lengths measured in the number of substitutions per site. This analysis involved 17 nucleotide sequences. All positions containing gaps and missing data were eliminated (complete deletion option). There was a total of 3,473 positions in the final data set. Evolutionary analyses were conducted in MEGA11.

and the transconjugant showed that the transconjugant not only acquired *catA2*, *tet(D)*, *dfrA14*, *aph(3′)-Ib*, *aph(6)-Id*, *qnrS1*, *sul2*, and *bla*$_{TEM-1}$ resistance genes but also emerged the corresponding resistance phenotype. The MIC of the ampicillin and tetracycline, enrofloxacin, ciprofloxacin, and ceftriaxone to the transconjugant were 16, eight, and two times, respectively, that of the recipient strain but the MIC of gentamicin was the same. The transconjugant was also sensitive to gentamicin (Table 3).

**TABLE 2** MIC of 18 antimicrobials to *E. hormaechei* M1

| Antimicrobials | MIC (μg/mL) | Results[a] |
|---|---|---|
| Ampicillin | >64 | R |
| Amoxicillin | >64 | R |
| Cefalexin | >64 | R |
| Ceftriaxone | 2 | I |
| Meropenem | 0.06 | S |
| Enrofloxacin | 4 | R |
| Ciprofloxacin | 2 | I |
| Chloramphenicol | >128 | R |
| Florfenicol | 16 | R |
| Oxytetracycline | >64 | R |
| Tetracycline | >64 | R |
| Doxycycline | 64 | R |
| Amoxicillin/clavulanate potassium | >32/16 | R |
| Trimethoprim/sulfamethoxazole | >4/76 | R |
| Sulfadiazine | >1024 | R |
| Gentamicin | 0.5 | S |
| Polymyxin | 0.06 | S |
| Fosfomycin | 256 | R |

[a]R, resistance; I, intermediate; S, sensitive.

**TABLE 3** Antimicrobials susceptibility determination of transconjugant and recipient stains

| Strains | Minimum inhibitory concn of antimicrobials (µg/mL) | | | | | | | | |
|---|---|---|---|---|---|---|---|---|---|
| | AMP[a] | SMZ | ACP | GEN | CHL | TET | ENR | CIP | CTRX |
| *E. coli* C600 | 4 | 0.25/4.75 | 16/8 | 0.5 | 2 | 4 | 0.25 | 1 | 0.25 |
| *E. coli* C600-pM1 | 64 | >4/76 | >64/32 | 0.5 | >128 | 64 | 2 | 2 | 0.5 |

[a]AMP, ampicillin; SMZ, trimethoprim-sulfamethoxazole; ACP, amoxicillin/clavulanate potassium; GEN,
 gentamicin; CHL, chloramphenicol; TET, tetracycline; ENR, enrofloxacin; CIP, ciprofloxacin; CTRX, ceftriaxone.

**Fitness cost and plasmid stability.** The transconjugant carrying *qnrS1*-positive plasmid originated from *E. hormaechei* M1 showed a relative fitness of 0.27 ± 0.047 (0 h VS. 24 h, $P < 0.01$). This result showed that the acquisition of *qnrS1*-bearing plasmid could place an energy burden on the bacterial host and incur fitness cost (Fig. 3A). Plasmid stability result showed that the transconjugant (*E. coli* C600-pM1) could be steadily passaged to 160 generations without antibiotic stress, then the plasmid was gradually lost. The plasmid containing rates in the 180 and 200; 220; 240; 260; 280; and 300 generations were 0.99 ± 0.019; 0.98 ± 0.019; 0.97 ± 0.033; 0.89 ± 0.011; 0.86 ± 0.019; and 0.81 ± 0.019, respectively. The plasmid carried by *E. hormaechei* M1 was not lost (Fig. 3B).

## DISCUSSION

The MDR *E. hormaechei* was reported in humans (28–31), animals (4–6, 32), and the environment (33, 34). The prevalence of MDR *Enterobacter* spp. carrying genes of extended-spectrum β-lactamases (ESBLs) and PMQR has been increasing worldwide (35). The isolation of MDR *E. hormaechei* from the cloaca of poultry has been reported (6). Furthermore, *E. hormaechei* strains have been isolated from animal feces (36). They suggested that fecal excretion is one of the reasons for the prevalence of MDR bacteria. The PMQR gene *qnrS* is prevalent in chicken feces (22, 37, 38). The *qnrS* gene was also found in the chicken feed (22). In this study, *E. hormaechei* carrying the *qnrS* gene was isolated from chicken feed. We speculate that it may result from the contamination caused by the exposure of the feed in the chicken farm to the environment. Previous study has shown that antibiotic-resistant bacteria carried in feed can transfer resistance genes to the gut microbiota after entering the zebrafish gut (39). Therefore, *E. hormaechei* carried in chicken feed may also transfer resistance genes to intestinal flora.

The isolation of MDR *E. hormaechei* from chicken feed has not been previously reported. This study firstly reported that *E. hormaechei* carrying *qnrS1* and *bla*TEM-1 genes was existed in chicken feed. The results of biochemical experiments were consistent with the description of *E. hormaechei*. The phenotypic-based tests were not accurate enough for species identification. Therefore, molecular identification was required (40).

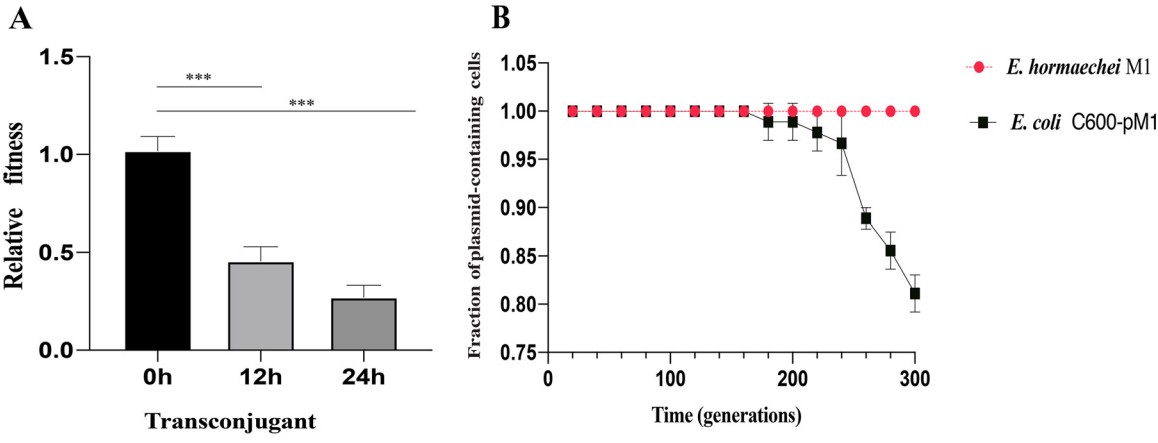

**FIG 3** The (A) figure shows relative fitness of transconjugant carrying *qnrS1*. A relative fitness of 1 indicates that the conjugant underwent no fitness cost. The conjugant showed a relative fitness of 0.27 ± 0.047 (0 h versus 24 h, $P < 0.01$) in this study. The (B) figure shows plasmid lost in serial passages without antibiotic selection pressure.

The MLST is a powerful tool for resolving the phylogeny of closely related species of the genus. The MLST especially suits to resolving complex phenotypes, such as virulence and antibiotic resistance in bacterial pathogens (41). The current reports about sequence types of *E. hormaechei* are ST78, ST873, ST66, ST419, ST145, ST50, ST118, and ST168 (42–44). However, the sequence type of *E. hormaechei* M1 was ST1736, which belonged to a new ST.

The *E. hormaechei* M1 carried IncF plasmid. Usually, IncF plasmids can encode several replicons, which is a kind of typical multireplicon IncF plasmid carrying the FII replicon together with FIA and FIB (45). IncF plasmid relaxase type is $MOB_F$, with the size from 45 to 200 kb, low copy number, conjugative, and most exist in Enterobacteriaceae (46). The functions of the proteins encoding the 40-kb *tra* operon were involved in formation of the F-type pilus (47). The conjugation transfer experiments and sequencing results also confirmed this point. The previous report showed that the *qnrS1* gene upstream element was IS2 (48). In this study, the *qnrS1* gene upstream element was IS2A which is a variant of IS2. More examples of a single IS mobilizing an adjacent region that includes one or more resistance genes are being identified, particularly in Gram-negative bacteria (49). In this investigation, we also found the IS407 and IS2A insertion elements were located upstream of $bla_{TEM-1}$ and *qnrS1* genes. Furthermore, the insertion elements upstream of antibiotic resistance genes (ARGs) affected antibiotic resistance phenotypes (50). The *E. hormaechei* M1 was low-level resistance to ciprofloxacin, which might be related to the upstream insertion element of *qnrS* gene. QnrS usually with $bla_{TEM}$ coexisted on the same plasmid (18, 51–53). In this research, the *qnrS1*, $bla_{TEM-1}$, *tet(D)*, *sul2*, *catA2*, *dfrA14*, *aph(3")-Ib*, and *aph(6)-Id* coexisted on pM1. Although pM1 harbored *aph(3")-Ib* and *aph(6)-Id* aminoglycosides resistance genes, it did show sensitive to gentamicin, which might be due to the mutations affecting the *aph(3")-Ib* and *aph(6)-Id* genes or their promoter region (54).

The plasmid stability experiment suggests that plasmid will be lost in the absence of antibiotic selection pressure. Similarly, the plasmid carrying the $bla_{NDM-5}$ gene was lost in *E. coli* by serial passage on medium without antibiotics (55), which may be attributed to non-selective conditions, for example, temperature and transcription, and replication and DNA topology that lead to plasmid instability (56). In addition, the competition between transcription and replication of the same DNA template leads to increased mutations and reduces the integrity of the genome due to replication and deletion events (57), which incurs plasmid loss. The previous report showed that the proportion of strains carrying plasmids with antibiotic resistance genes were decreased in absence selective pressure (58). Our research also found that the proportion of transconjugant (*E. coli* C600-pM1) was decreased in competition experiments. When the cost of plasmid carriage outweighs its benefit, plasmid-free proportion are expected to outcompete plasmid-carrying bacteria, eventually leading to plasmid loss (59). Plasmid maintenance is considered a metabolic burden to the host bacteria (60). Similarly, the transconjugant carrying pM1 showed a decreased fitness. Antibiotics may be a factor in maintaining the long-term stable existence of resistant plasmids.

The *E. hormaechei* strain carrying pC44-01 and pC4-001 was isolated from clinical samples of an Australian hospital (61). The *E. hormaechei* strain WCHEH020038 carrying p2_020038 was isolated from Center of Infectious Diseases, West China Hospital, Sichuan University. The *E. hormaechei* strain E70 carrying pE70 was isolated from a patient's urine sample. Previous reports show that the isolation of *E. hormaechei* are mainly focused on clinical samples (2, 62, 63). In this study, the *E. hormaechei* strain M1 strain was isolated from chicken feed, which is a new source. The comparison of IncF plasmids showed that pC44-01 and pC4-001 had higher similarity with pM1 but lacked encoding fimbriae genes. The p2_020038, pE70, pC44-01, pC4-001, and pM1 common regions encoded samB (DNA damage repair), sopA, and sopB (plasmid partitioning related genes); resD (resolvase); copper and silver cation efflux; IncFII and IncFIB plasmid replicons; and *traX* (transfer) genes, which could define the general IncF backbone. It was worth studying the functions of virulence factors located on pM1, such as, cation

efflux system; sopA and sopB; VagC; and fimbriae and hok toxic protein. The cation efflux system is related to virulence and might contribute to bacteria in environmental persistence and host colonization. Furthermore, bacterial influx systems for essential trace cations are known to contribute to pathogenesis (64–66). The VagC protein has been indirectly implicated in plasmid maintenance. The VagCD proteins were proved as functional TA systems with VagD the toxin and VagC its antitoxin (67, 68). The pM1 also carried the *VagCD* genes that may contribute to pathogenicity of *E. hormaechei*. Fimbriae act as independent virulence factors by promoting the establishment of bacteria and the innate host response, which is responsible for symptoms and tissue damage (69). The pM1 had more fimbriae encoding genes than other plasmids. It is possible that *E. hormaechei* M1 is highly pathogenic to the host. The *hok/sok* locus is a type I toxin/antitoxin plasmid stability element that increases bacterial tolerance to $\beta$-lactam antibiotics, which enhances bacteria survivability and pathogenicity in stressful growth conditions (70). These virulence factors allow bacteria to survive better under stress conditions. We also need to further explore the functions of these virulence factors to better reveal the pathogenic mechanism of *E. hormaechei*. In conclusion, MDR *E. hormaechei* in feed might be a risk control point of transmission of resistance genes. It is necessary to continuously strengthen the monitoring of MDR bacteria in feed and take effective measures to eliminate resistance genes and plasmids, so as to prevent the spread of resistance genes from aggravating clinical drug resistance.

**Conclusion.** This is the first report that *E. hormaechei* carrying *qnrS* and $bla_{\text{TEM-1}}$ has been isolated from chicken feed. It is a MDR *E. hormaechei* strain. Importantly, the plasmid pM1 not only contained multiple resistance genes but also virulence factors related to its pathogenicity. The concern is that chicken feed and poultry products could serve as a vehicle for these MDR bacteria, which could be transferred between animals and humans through the food chain. The IncF of conjugation plasmids could transfer intergenus. If pathogenic bacteria acquire the drug-resistant plasmids, it will lead to the failure of clinical treatment. The MDR *E. hormaechei* may be a public health issue. There is an urgent need for close epidemiologic surveillance to control their further spread.

## MATERIALS AND METHODS

**Isolation and identification of strains.** The *E. hormaechei* strain M1 was isolated from feed of a chicken farm in Hubei Province, China. The feed samples were added to 4 mL LB broth (Hopebio, Qingdao, China) and incubated at 37°C for 4 h to 5 h, then 100-$\mu$L cultures were spread on the LB agar plates supplemented with enrofloxacin (1 $\mu$g/mL) overnight at 37°C. The single colonies were randomly selected from the LB agar plates and identified by 16S rRNA sequencing. Meanwhile, the *E. hormaechei* strain M1 was identified by conventional biochemical tests.

**Reagents.** The standards of enrofloxacin, amoxicillin, tetracycline, chloramphenicol, ampicillin, trimethoprim, florfenicol, doxycycline, gentamicin, oxytetracycline, sulfadiazine, and sulfamethoxazole were purchased from Dr. Ehrenstorfer (Germany). Clavulanate potassium, meropenem, polymyxin, fosfomycin, cefalexin, and ceftriaxone were obtained from China National Institutes for Drug Control. Rifampicin was purchased from Shanghai Yuanye Bio-Technology Co., Ltd. Ciprofloxacin standards was purchased by MedChemExpress (New Jersey, USA). The 2 × EasyTaq PCR Master Mix (Dye plus) and Phanta super-fidelity DNA polymerase were purchased from Vazyme (Nanjing, China). Bacterial Micro Biochemical Tubes were bought from Hangzhou Microbial Regent Co., Ltd and Hopebio, Qingdao, China.

**Whole genome sequencing and molecular analysis of E. hormaechei M1.** *E. hormaechei* strain M1 was collected by propagating culture in LB broth (Hopebio, Qingdao, China), centrifuged and frozen in liquid nitrogen. Whole genome sequencing (WGS) was done by Oxford Nanopore Technologies DNA sequencing platform (Biomarker Technologies, Beijing, China). The genes were automatically annotated by Prodigal v2.6.3. Then manual inspection and correction used the BLASTn and BLASTp programs (https://blast.ncbi.nlm.nih.gov/Blast.cgi). Plasmid type comparison of PlasmidFinder was carried out on (http://www.genomicepidemiology.org/) and the resistance genes encoded proteins were blasted by Uniprot (https://www.uniprot.org/). Alignments of similar IncF plasmids were created by BRIG tools (https://sourceforge.net/projects/brig/). The accession numbers of the plasmids were as follows: p2-020038 (CP031723.1), pE70(CP046273.1), pC44-01 (CP042567.1), pC4-001 (CP042541.1), and pM1 (CP090910).

**Phylogenetic tree.** The *E. hormaechei* M1 (accession number CP090909) genome was uploaded onto MLST (https://pubmlst.org/). Then the *dnaA, fusA, gyrB, leuS, pyrG, rplB,* and *rpoB* housekeeping genes were determined (23). Taxonomic evaluation of the *E. hormaechei* based on MLST. The phylogenetic tree of

17 strains of *E. hormaechei* was produced by MEGA11 (https://www.megasoftware.net/). The accession numbers of these strains were as follows: DSM 16691 (CP017179.1), DSM16687 (CP017180.1), L51 (CP033102.1), RHBSTW-00012 (CP058191.1), A1 (CP031574.1), 14102KSN (CP045312.1), LMG27195 (CP017183.1), Ec61 (CP053103.1), DSM14563 (CP017186.1), Eh1 (CP034754.1), OIPH-N069 (AP019817.1), FDAARGOS_1533 (CP083613.1), ME-1 (CP041733.1), and BD-50-Eh (CP063224.1) strains were isolated from humans. Moreover, the C15117 (CP032841.1) and CM18-216 (CP050311.1) stains were isolated from hospital environment. Their genomes were obtained by NCBI (https://www.ncbi.nlm.nih.gov/).

**Susceptibility tests.** The MICs of ampicillin, amoxicillin, cefalexin, enrofloxacin, ciprofloxacin, chloramphenicol, florfenicol, tetracyclines, oxytetracycline, tetracycline, doxycycline, sulfadiazine, fosfomycin, camoxicillin/clavulanate potassium, trimethoprim-sulfamethoxazole, meropenem, gentamicin, and polymyxin were measured by broth microdilution and agar dilution methods according to the recommendations of the CLSI document M100-30th ed (CLSI 2020) (24). According to the criteria of Enterobacteriaceae, the *E. coli* ATCC25922 strain was used as quality control strain. The susceptibility tests were repeated three times.

**Conjugation transfer experiment.** MacConkey plates were supplemented with 125 $\mu$g/mL rifampicin for counting recipient bacteria, and with 125 $\mu$g/mL rifampicin and 2 $\mu$g/mL enrofloxacin for counting the conjugants. *E. hormaechei* M1 and *E. coli* C600 were used as donor and recipient strain, respectively. They were subcultured on eosin methylene blue (EMB) plates, and a single colony was selected and inoculated into 1 mL LB broth and cultured with shaking at 37°C for 5 h to 6 h. The 0.22-$\mu$m filter membrane was placed on the antibiotic-free LB agar medium. Then, 20 $\mu$L of the donor bacteria and 60 $\mu$L of the recipient bacteria were mixed in an EP tube, which were equally aliquoted onto the filter membrane and incubated overnight at 37°C. The bacterial lawn was washed down with 1 mL of antibiotic-free LB broth, the 100-$\mu$L bacterial solution was proceeded with multiple dilution. Bacterial solution with a dilution of 101:1,010 was spread on the LB agar plates supplemented with 125 $\mu$g/mL rifampicin and 2 $\mu$g/mL enrofloxacin, then the plates were selected from the colonies between 30 and 300 for counting. The bacterial solution with a dilution of 101:1,010 was spread on the LB agar supplemented with 125 $\mu$g/mL rifampicin plates respectively, and the plates were selected from the colonies between 30 and 300 for counting. The conjugation transfer efficiency $=$ number of transconjugants/number of recipients. Then, five conjugants were randomly selected from the LB agar plates supplemented with 125 $\mu$g/mL rifampicin and 2 $\mu$g/mL enrofloxacin, and their homology with the recipient strain *E. coli* C600 were verified by ERIC-PCR.

**Plasmid stability tests.** To estimate plasmid stability, *E. coli* C600 conjugants and *E. hormaechei* M1 carrying qnrS were cultured in antibiotic-free LB broth, respectively, overnight at 37°C. The cultures were diluted 1:10³ in fresh LB medium and were further incubated overnight at 37°C. Serial passaging of 1 $\mu$L of overnight culture to 1 mL of LB was performed daily, approximately 10 generations per passage. Every 20 generations, the cultures were diluted and spread on LB agar plates. The ratio of colonies grown on antibiotic-supplemented LB agar (2 $\mu$g/mL enrofloxacin) compared with that on antibiotic-free LB agar was determined in triplicate. Presence of qnrS gene in the host after each passage was verified by PCR. The colonies grown on antibiotic-free/supplemented agar were randomly selected (~30 colonies per agar) as DNA template (25).

**Competition experiments to assess *in vitro* fitness.** To assess the fitness impact of qnrS gene carriage, pairwise competition assays were carried out using the *E. coli* C600 transconjugants carrying qnrS gene competed with its plasmid-free counterparts. As described previously, 24 h competition experiments were performed (26). Briefly, cultures were adjusted to a 1.0 McFarland standard, which were diluted 1:10⁴ and then mixed at a volumetric ratio of 1:1 (time point zero). Colony counts were determined by plating serial dilutions of mixed cultures on LB agar (LBA) with and without enrofloxacin (2 $\mu$g/mL) at 0 h, 12 h, and 24 h. The number of CFU growing on antibiotic-supplemented LBA was subtracted from the number of CFU growing on antibiotic-free LBA to determine the number of susceptible cells in the mixed population. All experiments were performed in triplicate and at least four replicates of each competition assay were performed. The relative fitness is calculated by the ratio of the growth rate of the resistant cells to that of the susceptible ones according to previous report (27). A relative fitness of 1 indicated that the transconjugants undergo no fitness cost, whereas a ratio of greater than or less than 1 indicated increased or decreased fitness, respectively.

**Statistical analysis.** Using GraphPad Prism 8.0 statistical software, the value was expressed by mean $\pm$ SD, and the difference between different time points was analyzed by one-way ANOVA. ***, $P \leq 0.01$ (extremely significant difference).

**Data availability.** The whole genome sequences of *E. hormaechei* strain M1 were uploaded into NCBI databases. Their accession numbers are as follows: *E. hormaechei* M1 (CP090909), pM1 (CP090910).

## ACKNOWLEDGMENTS

We declare no conflict of interest.

This work was supported by the National Key R&D Program of China (grant number 2017YFC1600100), Walmart Foundation (Project # 61626817) supported by Walmart Food Safety Collaboration Center, and the National Natural Science Foundation of China (NSFC) (31772736).

M.D. conceived and designed the study, and wrote, reviewed, and edited the manuscript. Z.C., L.C., Q.L., Y.Z., K.G., T.H., and F.Z. performed the experiments; Z.C., L.C., and Q.L. wrote the draft manuscript. All authors participated in the interpretation of the results and read and approved the manuscript.

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
