## [Reviewer comments · Microbiology Spectrum]

Microbiology Spectrum

Phenotypic and genotypic characterization of multi-drug resistant *Enterobacter hormaechei* carrying *qnrS* gene isolated from chicken feed in China

Zhengzheng Cao, Luqing Cui, Quan Liu, Fangjia Liu, Yue Zhao, Kaixuan Guo, Tianyu Hu, Fan Zhang, Xijing Sheng, Xiangru Wang, Zhong Peng, and Menghong Dai

Corresponding Author(s): Menghong Dai, 1The Cooperative Innovation Center for Sustainable Pig Production, Huazhong Agricultural University, Wuhan 430070, China 2MOA Key Laboratory of Food Safety Evaluation/National Reference Laboratory of Veterinary Drug Residue (HZAU), Huazhong Agricultural University, Wuhan 430070, China

Review Timeline:

Submission Date:	December 6, 2021
Editorial Decision:	January 11, 2022
Revision Received:	February 24, 2022
Editorial Decision:	March 20, 2022
Revision Received:	March 22, 2022
Accepted:	March 27, 2022

Editor: Kapil Chousalkar

Reviewer(s): The reviewers have opted to remain anonymous.

Transaction Report:

DOI: <https://doi.org/10.1128/spectrum.02518-21>

January 11, 2022

Prof. Menghong Dai

1The Cooperative Innovation Center for Sustainable Pig Production, Huazhong Agricultural University, Wuhan 430070, China
2MOA Key Laboratory of Food Safety Evaluation/National Reference Laboratory of Veterinary Drug Residue (HZAU), Huazhong Agricultural University, Wuhan 430070, China
Shizishan Street Number 1
Wuhan
China

Re: Spectrum02518-21 (Phenotypic and genotypic characterization of multi-drug resistant *Enterobacter hormaechei* carrying qnrS gene isolated from chicken feed in China)

Dear Prof. Menghong Dai:

Thank you for submitting your manuscript to Microbiology Spectrum. Your paper was reviewed by two experts and both of them have suggested major modifications in your paper. Both of them have specifically suggested extensive and professional editing/proof-reading" to improve the quality of your paper. When submitting the revised version of your paper, please provide (1) point-by-point responses to the issues raised by the reviewers as file type "Response to Reviewers," not in your cover letter, and (2) a PDF file that indicates the changes from the original submission (by highlighting or underlining the changes) as file type "Marked Up Manuscript - For Review Only". Please use this link to submit your revised manuscript - we strongly recommend that you submit your paper within the next 60 days or reach out to me. Detailed instructions on submitting your revised paper are below.

Link Not Available

Sincerely,

Kapil Chousalkar

Journals Department
Reviewer comments:

Reviewer #1 (Comments for the Author):

This manuscript reports a series of laboratory experiments and genomic analysis that partially characterized an *Enterobacter hormaechei* strain recently isolated from chicken feed. The key finding of this report is the identification of MDR and the presence of the qnrS gene in a pM1 plasmid. For the most part this is a relatively relevant topic and the research follow a relatively straight forward approach of AMR phenotypic and genotypic characterization.

There are a number of issues that should be addressed. Because of multiple grammar errors, authors are recommended to thoroughly proofread it and edit accordingly.

L52 - First sentence is redundant merge with second sentence to read "The *Enterobacter hormaechei* is a member of the E. cloacea complex and has different subspecies"

L54 - insert "the" before "environment"

L55 - change "caused" to "causes"
L56 - insert "a" before "green"
L57-59 - sentence does not make sense, please re-write.
L59 - delete "gene"
L60 - insert "a" before "protein"
L61 - quinolones should be singular
L61 - split paragraph before "PMQR"
L65 - correct spelling of "hormaechei"
L68-81 - this is a summary of the work conducted in this report and it should be deleted because it is redundant with the M&M section. Instead, the objectives of the study should be briefly stated.
L85 - insert a space in "4ml" and change "ml" to "mL" to be consistent with the rest of the manuscript.
L86 - delete "the feed"
L102 - "Molecular" does not need to be capitalized
L104 - change "expanding" to "propagating"
L121-128 - is this the list of the 17 strains? What do you mean by "were Homo sapiens"?
L129 - how many times the susceptibility tests were repeated?
L136 - insert "strain" before "was"
L139 - delete "The"
L140 - replace "the MacConkey plates were supplemented" with "and"
L142 - delete first "strain" after "donor"
L143 - delete "an"
L145 - insert a space in "0.22 μ m"
L149 - "proceed" - did you mean "processed"?
L166 - insert a space in "2 μ g"
L191 - Result should be plural
L234 - insert "was sensitive" before "to"
L235 - delete "was sensitive"
L242-243 - what do you mean "but also emerged corresponding resistance"
L252 - revise sentence "passage" is a noun not a verb
L256 - revise sentence, poor grammar.
L258 - delete "mainly" it is unnecessary
L258-263 - revise paragraph, only the first two sentences are related. The last sentence is not necessary a conclusion from the previous 3 sentences. It needs better logic.
L264- a single detection on a chicken feed sample can not lead to the conclusion that it is an emergence. Because there is no information about the representativeness of the feed sample, all that can be concluded is that this is the first isolation from chicken feed.
L337 - revise sentence, poor grammar.
Reference section - needs to be thoroughly edited to comply with a consistent referencing format. All bacterial names are not italicized.
Table 1 - the results of a positive or standard *E. hormaechei* are missing in this table and in its experimental design.
Table 2 - delete since all of those results are reported in the text

Reviewer #2 (Comments for the Author):

This study by Cao et al, summarizes the Phenotypic and genotypic characterization of a MDR *Enterobacter hormaechei* isolate carrying *qnrS* gene isolated from chicken feed. While this study sheds some light into the emergence of Plasmid-mediated quinolone resistance (PMQR) genes in newer *Enterobacter* spp. isolated from Cattle feeds, it falls short in demonstrating the linkage between the origins of the PMQR in Chicken Feed and their spread in Chickens.

Abstract

Lines 18, 45, 59- Plasmid-mediated "quinolones" resistance (PMQR) gene- Could avoid plural for quinolone while mentioning about PMQR gene resistance (PMQR). (could be- Plasmid-mediated quinolone resistance (PMQR) gene)

Line 25- Taxonomic of the *E. hormaechei* based on multi-locus sequence type- Grammatical error (could rephrase as; "Taxonomy of the *E. hormaechei* was based on multi-locus sequence type..")

Importance

Lines 42-50- This paragraph has information described in bits and pieces. The authors could rephrase this section to improve the scientific coherence of this study.

Introduction

Lines 64-65- However, there are few reports on the prevalence of qnrS in *E. hormaechei*. No Reference has been provided for this statement.

In fact, the authors have failed to quote/refer/acknowledge another recently published study by Ai et al, reporting the co-existence of qnr genes in a MDR *E. hormaechei* isolated from humans.

(<https://www.frontiersin.org/articles/10.3389/fmicb.2021.676113/full>)

Lines 66-81- This portion sounds like a rehash of their Abstract, with a summary of their Abstract/results (verbatim from the Abstract on several instances). Requires extensive rewriting (with minimal repetition) to setup the context of this study in this section.

Methodology

Line- 87- Why a differential media such as EMB/MaConkey agar plates were not used here (in addition to a general media- LB agar)?

Line 92-94- Why were Antibiotics purchased from a private entity (Dr. Ehrenstorfer (Germany)) and not a commercial vendor following standard Quality procedures.

Line 126- "BD-50-Eh (CP063224.1) were *Homo sapiens*"- This section is not clear- Unable to deduce what the authors are trying to convey here.

Line 147- "which were equably dripped on the filter membrane"- could be written as "which were equably aliquoted onto the filter membrane"

Inconsistent "verb tense" throughout the methodology section, that swiches back and forth.

Results

Sections 3-1 & 3.2- The authors have sequenced the Whole genome and Plasmids of *Enterobacter hormaechei*, but have not provided a Genbank ID for the sequences obtained from this study in these sections (unless I have overlooked).

Section 3.3- A table with a list of "Unique genes in pM1" (as compared to other plasmids in Fig 1) would be helpful for easier Interpretation of the data.

Lines 216-217- Were there sequence evidence for a Operon? If so, did the authors try using a PCR for amplifying this Operon region among Transconjugants?

Lines 234-235- However, *E. hormaechei* M1 to meropenem, gentamicin and polymyxin was sensitive- Could be rephrased as "However, *E. hormaechei* M1 was sensitive to meropenem, gentamicin and polymyxin"

Discussion

Line 275- "size from 45 to 200"- what is the "unit" here- 45-200 kb?

Lines 282-284- Consider rephrasing this sentence- "In this research also found IS407 and IS2A insertion elements located upstream of blaTEM-1 and qnrS1 genes. Insertion elements upstream of genes affect antibiotic resistance phenotypes (44)"

"Chicken Feed" being the epicenter of this study, No discussion points have been made or reference(s) provided to discuss the origins of the PMQR in Chicken Feed?

Staff Comments:

Preparing Revision Guidelines

Please return the manuscript within 60 days; if you cannot complete the modification within this time period, please contact me. If you do not wish to modify the manuscript and prefer to submit it to another journal, please notify me of your decision immediately so that the manuscript may be formally withdrawn from consideration by Microbiology Spectrum.

Responses to Reviewers Comments

We revised the article and highlighted it with a yellow background.

Reviewer #1 (Language comments to the Authors)

Response: Thanks a lot for your comments. We modified the tense and language thoroughly. Hope this version will be better.

1. L52 - First sentence is redundant merge with second sentence to read "The *Enterobacter hormaechei* is a member of the *E. cloacea* complex and has different subspecies"

Response: Thanks for your comments. We revised and marked it with yellow background (Line 59-60).

2. L54 - insert "the" before "environment"

Response: Thanks for your comments. We inserted "the" before "environment" (Line 60).

3. L55 - change "caused" to "causes"

Response: Thanks for your comments. We replaced "caused" by "causes" (Line 61)

4. L56 - insert "a" before "green"

Response: Thanks for your comments. We inserted "a" before "green" (Line 63)

5. L57-59 - sentence does not make sense, please re-write.

Response: Thanks for your comments. We revised and marked it with yellow background (Line 63-65).

6. L59 - delete "gene"

Response: Thanks for your comments. We deleted "gene" (Line 65)

7. L60 - insert "a" before "protein"

Response: Thanks for your comments. We inserted "a" before "protein"(Line 66)

8. L61 - quinolones should be singular

Response: Thanks for your comments. We replaced "quinolones" by "quinolone" (Line 66)

9. L61 - split paragraph before "PMQR"

Response: Thanks for your comments. We revised it. (Line 69)

10. L65 - correct spelling of "hormaechei"

Response: Thanks for your comments. We replaced "*hormaechei*" by "*hormaechei*" (Line 72)

11. L68-81 - this is a summary of the work conducted in this report and it should be deleted because it is redundant with the M&M section. Instead, the objectives of the study should be briefly stated.

Response: Thanks for your comments. We deleted it.

12. L85 - insert a space in "4ml" and change "ml" to "mL" to be consistent with the rest of the

manuscript.

Response: Thanks for your comments. We replaced “4ml” by “4 mL” (Line 87)

13. L86 - delete "the feed"

Response: Thanks for your comments. We deleted “the feed” (Line 88)

14. L102 - "Molecular" does not need to be capitalized

Response: Thanks for your comments. We replaced “Molecular” by “molecular” (Line 104)

15. L104 - change "expanding" to "propagating"

Response: Thanks for your comments. We replaced “expanding” by “propagating” (Line 106)

16. L121-128 - is this the list of the 17 strains? What do you mean by "were Homo sapiens"?

Response: Thanks for your comments. We listed the accession numbers of 17 strains. The meaning of “were Homo sapiens” is “ These strains were isolated from humans” (Line 123-129)

17. L129 - how many times the susceptibility tests were repeated?

Response: Thanks for your comments. We repeated this experiment three times.

18. L136 - insert "strain" before "was"

Response: Thanks for your comments. We inserted “strain” before “was”.(Line 139)

19. L139 - delete "The"

Response: Thanks for your comments. We deleted it. (Line 142)

20. L140 - replace "the MacConkey plates were supplemented" with "and"

Response: Thanks for your comments. We replaced “the MacConkey plates were supplemented” by “and”. (Line 143)

21. 142 - delete first "strain" after "donor"

Response: Thanks for your comments. We deleted it. (Line 144-145)

22. L143 - delete "an"

Response: Thanks for your comments. We deleted it. (Line 145)

23. L145 - insert a space in "0.22µm"

Response: Thanks for your comments. We replaced “0.22µm” by “0.22 µm”. (Line 147)

24. L149 - "proceed" - did you mean "processed"?

Response: Thanks for your comments. We replaced “proceed” by “proceeded”. (Line 151)

25. L166 - insert a space in "2µg"

Response: Thanks for your comments. We replace “2µg” by “2 µg”. (Line 168)

26. L191 - Result should be plural

Response: Thanks for your comments. We replaced “Result” by “Results”. (Line 192)

27. L234 - insert "was sensitive" before "to"

Response: Thanks for your comments. We inserted “was sensitive” before the“to”. (Line 235)

28. L235 - delete "was sensitive"

Response: Thanks for your comments. We deleted it.

29. L242-243 - what do you mean "but also emerged corresponding resistance"

Response: Thanks for your comments. We replaced “but also emerged corresponding resistance” by “but also emerged the corresponding resistance phenotype”. (Line 243)

30. L252 - revise sentence "passage" is a noun not a verb

Response: Thanks for your comments. We “passage” by “passaged”. (Line 252)

31. L256 - revise sentence, poor grammar.

Response: Thanks for your comments. We revised to “The plasmid carried by *E. hormaechei* M1 was not lost.” (Line 255-256)

32. L258 - delete "mainly" it is unnecessary

Response: Thanks for your comments. We deleted it. (Line 258)

33. L258-263 - revise paragraph, only the first two sentences are related. The last sentence is not necessary a conclusion from the previous 3 sentences. It needs better logic.

Response: Thanks for your comments. We revised and marked it with yellow background. (Line 258-263)

34. L264- a single detection on a chicken feed sample can not lead to the conclusion that it is an emergence. Because there is no information about the representativeness of the feed sample, all that can be concluded is that this is the first isolation from chicken feed.

Response: Thanks for your comments. We revised and marked it with yellow background. (Line 271)

35. L337 - revise sentence, poor grammar.

Response: Thanks for your comments. We revised and marked it with yellow background.(Line 353-354)

36. Reference section - needs to be thoroughly edited to comply with a consistent referencing format.

Response: Thanks for your comments. We adopted the ASM journal reference format.

37. All bacterial names are not italicized.

Response: Thanks for your comments. We changed all bacterial names to italics.

38. Table 1 - the results of a positive or standard *E. hormaechei* are missing in this table and in

its experimental design.

Response: Thanks for your comments. The results of biochemical identification were according with the reports in the reference. In addition, the results of whole genome sequencing can also confirm that the strain is *Enterobacter hormaechei*. So we did not perform biochemical identification of standard strains.

39. Table 2 - delete since all of those results are reported in the text

Response: Thanks for your comments. We deleted Table 2 in the text.

Reviewer #2

1. Lines 18, 45, 59- Plasmid-mediated "quinolones" resistance (PMQR) gene- Could avoid plural for quinolone while mentioning about PMQR gene resistance (PMQR). (could be Plasmid-mediated quinolone resistance (PMQR) gene)

Response: Thanks for your comments. We revised and marked it with yellow background.(Line 19, 44, 66)

2. Line 25- Taxonomic of the *E. hormaechei* based on multi-locus sequence type- Grammatical error (could rephrase as; "Taxonomy of the *E. hormaechei* was based on multi-locus sequence type.")

Response: Thanks for your comments. We replaced "Taxonomic of the *E. hormaechei* based on multi-locus sequence type" by "Taxonomy of the *E. hormaechei* was based on multilocus sequence typing". (Line 26)

3. Lines 42-50- This paragraph has information described in bits and pieces. The authors could rephrase this section to improve the scientific coherence of this study.

Response: Thanks for your comments. We rephrased it. (Line 43-56)

4. Lines 64-65- However, there are few reports on the prevalence of *qnrS* in *E. hormachei*. No Reference has been provided for this statement.

Response: Thanks for your comments. We provided relative reference. (Line 72)

5. Lines 66-81- This portion sounds like a rehash of their Abstract, with a summary of their Abstract/results (verbatim from the Abstract on several instances). Requires extensive rewriting (with minimal repetition) to setup the context of this study in this section.

Response: Thanks for your comments. We rephrased it. (Line 76-83)

6. Line- 87- Why a differential media such as EMB/MaConkey agar plates were not used here (in additon to a general media- LB agar)?

Response: Thanks for your comments. More bacterial species could be grown on LB plates than on selective media. We isolated the bacteria carrying the *qnrS* gene from the chicken

feed. Because the species of the strain carrying the *qnrS* gene was unknown. Then, the resistant bacteria on the plates required to be identified by 16s rRNA sequencing. Therefore, the LB plates were used.

7. Line 92-94- Why were Antibiotics purchased from a private entity (Dr. Ehrenstorfer (Germany)) and not a commercial vendor following standard Quality procedures.

Response: Thanks for your comments. Dr. Ehrenstorfer's antibiotic standards are of higher purity. The experimental results are more accurate and reliable.

8. Line 126- "BD-50-Eh (CP063224.1) were Homo sapiens"- This section is not clear- Unable to deduce what the authors are trying to convey here.

Response: Thanks for your comments. We revised it. (Line 124-129)

9. Line 147- "which were equably dripped on the filter membrane"- could be written as "which were equably aliquoted onto the filter membrane"

Response: Thanks for your comments. We replaced "which were equably dripped on the filter membrane" by "which were equably aliquoted onto the filter membrane". (Line 149-150)

10. Inconsistent "verb tense" throughout the methodology section, that switches back and forth.

Response: Thanks for your comments. We modified the verb tense thoroughly. Hope this version will be better.

11. Sections 3-1 & 3.2- The authors have sequenced the Whole genome and Plasmids of *Enterobacter hormaechei*, but have not provided a Genbank ID for the sequences obtained from this study in these sections (unless I have overlooked).

Response: Thanks for your comments. We provided the Genbank ID of Whole genome and Plasmid of *Enterobacter hormaechei* M1 in text. (Line 117, 119)

12. Section 3.3- A table with a list of "Unique genes in pM1" (as compared to other plasmids in Fig 1) would be helpful for easier Interpretation of the data.

Response: Thanks for your comments. We added a new table 2 in the text. (Line 209,650)

13. Lines 216-217- Were there sequence evidence for a Operon? If so, did the authors try using a PCR for amplifying this Operon region among Transconjugants?

Response: Thanks for your comments. The operon sequence (pM1-IS407-blaTEM-1-IS2A-qnrS1-ISKra4-tnpR (95,426 .. 102,007).gb) . These elements we had compared through NCBI were correct. The plasmid was extracted from the transconjugants. Then the antibiotics resistance genes, replicons, insertion elements and transposable elements on the plasmid were amplified and verified by PCR.

14. Lines 234-235- However, *E. hormaechei* M1 to meropenem, gentamicin and polymyxin was sensitive- Could be rephrased as "However, *E. hormaechei* M1 was sensitive to meropenem, gentamicin and polymyxin"

Response: Thanks for your comments. We replaced “However, *E. hormaechei* M1 to meropenem, gentamicin and polymyxin was sensitive” by “However, *E. hormaechei* M1 was sensitive to meropenem, gentamicin and polymyxin” (Line 235)

15. Line 275- "size from 45 to 200"- what is the "unit" here- 45-200 kb?

Response: Thanks for your comments. We replaced “size from 45 to 200” by “with the size from 45 to 200 kb”. (Line 285)

16. Lines 282-284- Consider rephrasing this sentence- "In this research also found IS407 and IS2A insertion elements located upstream of *bla*_{TEM-1} and *qnrS1* genes. Insertion elements upstream of genes affect antibiotic resistance phenotypes (44)"

Response: Thanks for your comments. We replaced “In this research also found IS407 and IS2A insertion elements located upstream of *bla*_{TEM-1} and *qnrS1* genes. Insertion elements upstream of genes affect antibiotic resistance phenotypes” by “In this investigation, we also found the IS407 and IS2A insertion elements were located upstream of *bla*_{TEM-1} and *qnrS1* genes. Furthermore, the insertion elements upstream of antibiotic resistance genes (ARGs) affected antibiotic resistance phenotypes”. (Line 292-295)

17. "Chicken Feed" being the epicenter of this study, No discussion points have been made or reference(s) provided to discuss the origins of the PMQR in Chicken Feed?

Response: Thanks for your comments. We discussed possible sources of the *qnrS* gene in chicken feed. (Line 263-267)

March 20, 2022

Dr. Menghong Dai

MOA Laboratory for Risk Assessment of Quality and Safety of Livestock and Poultry Products, Huazhong Agricultural University
Wuhan 430070
China

Re: Spectrum02518-21R1 (Phenotypic and genotypic characterization of multi-drug resistant *Enterobacter hormaechei* carrying qnrS gene isolated from chicken feed in China)

Dear Dr. Menghong Dai:

Thank you for submitting the revision. Both reviewers suggested a minor revision. If you are prepared to make minor revision, I am prepared to consider your manuscript further.

Link Not Available

Sincerely,

Kapil Chousalkar

Journals Department
Reviewer comments:

Reviewer #1 (Comments for the Author):

Many of the recommendations were incorporated, despite the authors assurances, several important changes were not actually performed.

L59 - sentence was not modified as suggested in the previous version for L52

L77-84 - paragraph was modified, but it still includes a summary of the work performed. It is inappropriate for an introduction.

Please delete and just state the objectives of the work.

L135-143 - please state that you repeated the measurement three times.

L268 - insert "strains" before "have"

Reference section - the recommendation of corrections to this section was ignored. Bacterial names are still not italicized and the format is not consistent.

Table 2 - the previous table 2 was deleted, but the new table 2 can also be reported in the text and the table deleted.

Reviewer #2 (Comments for the Author):

Minor Typos

Line 44- To add "an"- Enterobacter hormaechei is an opportunistic pathogen.

Line 74 and 466-469- Reference 21- The reference cited is incomplete- It should read as "First Report of Coexistence of blaSFO-1 and blaNDM-1 β -Lactamase Genes as Well as Colistin Resistance Gene mcr-9 in a Transferrable Plasmid of a Clinical Isolate of Enterobacter hormaechei"

Staff Comments:

Preparing Revision Guidelines

Please return the manuscript within 60 days; if you cannot complete the modification within this time period, please contact me. If you do not wish to modify the manuscript and prefer to submit it to another journal, please notify me of your decision immediately so that the manuscript may be formally withdrawn from consideration by Microbiology Spectrum.

Response to Reviewers Comments

Thanks for your comments. We revised the article and highlighted it with a yellow background.

Reviewer #1

1. L59 - sentence was not modified as suggested in the previous version for L52

Response: Thanks for your comments. We revised and marked it with yellow background (Line 58-59).

2. L77-84 - paragraph was modified, but it still includes a summary of the work performed. It is inappropriate for an introduction. Please delete and just state the objectives of the work.

Response: Thanks for your comments. We revised and marked it (Line 75-78)

3. L135-143 - please state that you repeated the measurement three times.

Response: Thanks for your comments. The susceptibility tests were repeated three times. (Line 135)

4. L268 - insert “strains” before “have”

Response: Thanks for your comments. We inserted “strains” before “have” (Line 260).

5. Reference section - the recommendation of corrections to this section was ignored. Bacterial names are still not italicized and the format is not consistent.

Response: Thanks for your comments. We revised and marked it with yellow background (Line 378-634).

6. Table 2 - the previous table 2 was deleted, but the new table 2 can also be reported in the text and the table deleted.

Response: Thanks for your comments. We deleted the table 2 and added the description in the text (Line 203-206).

Reviewer #2

1. Line 44- To add “an”- *Enterobacter hormaechei* is an opportunistic pathogen.

Response: Thanks for your comments. We added “an” in “*Enterobacter hormaechei* is an opportunistic pathogen”(Line 43)

2. Line 74 and 466-469- Reference 21- The reference cited is incomplete. It should read as "First Report of Coexistence of *bla*_{SFO-1} and *bla*_{NDM-1} β -Lactamase Genes as Well as Colistin Resistance Gene *mcr-9* in a Transferrable Plasmid of a Clinical Isolate of *Enterobacter hormaechei*"

Response: Thanks for your comments. We revised it (Line 451-455).

March 27, 2022

Prof. Menghong Dai

1The Cooperative Innovation Center for Sustainable Pig Production, Huazhong Agricultural University, Wuhan 430070, China
2MOA Key Laboratory of Food Safety Evaluation/National Reference Laboratory of Veterinary Drug Residue (HZAU), Huazhong Agricultural University, Wuhan 430070, China
Shizishan Street Number 1
Wuhan
China

Re: Spectrum02518-21R2 (Phenotypic and genotypic characterization of multi-drug resistant *Enterobacter hormaechei* carrying qnrS gene isolated from chicken feed in China)

Dear Prof. Menghong Dai:

Your manuscript has been accepted, and I am forwarding it to the ASM Journals Department for publication. You will be notified when your proofs are ready to be viewed.

Sincerely,

Kapil Chousalkar
Editor, Microbiology Spectrum
